# Vaccines Administration in the Perspective of Patient Safety and Quality of Healthcare: Lesson from the Experience of an Italian Teaching Hospital for Pandemic Preparedness

**DOI:** 10.3390/vaccines10091495

**Published:** 2022-09-08

**Authors:** Francesco De Micco, Anna De Benedictis, Lorenzo Sommella, Andrea Di Mattia, Laura Leondina Campanozzi, Rossana Alloni, Vittoradolfo Tambone

**Affiliations:** 1Bioethics and Humanities Research Unit, Campus Bio-Medico University of Rome, 00128 Rome, Italy; 2Department of Clinical Affairs, Campus Bio-Medico University Hospital Foundation, 00128 Rome, Italy; 3Nursing Sciences Research Unit, Campus Bio-Medico University of Rome, 00128 Rome, Italy; 4Department of Medical Affairs, Campus Bio-Medico University Hospital Foundation, 00128 Rome, Italy

**Keywords:** COVID-19 vaccination, healthcare risk management, healthcare quality, immunization center, clinical governance, hospital-based vaccination

## Abstract

The development and administration of vaccines against COVID-19 was a key element in the fight against the pandemic, as it protected health systems and helped restore global economies. National implementation plans and vaccination strategies for COVID-19 vaccines ensured the immunization of large segments of the population in the shortest time. However, even before the start of the vaccination campaign, it was clear to decision-makers that the usual methods of vaccination were not suitable. The aim of this report is to share the experience of an Italian teaching hospital in the organisation of spaces and activities of healthcare workers to realise a safe vaccination campaign. An in-depth analysis of how the vaccination campaign was organised could be useful to understand strengths and weaknesses learnt from this experience and plan an effective, efficient, and resilient response to future pandemics right away. The adoption of a systemic clinical risk management (SCRM) could guarantee healthcare organizations a more adequate and resilient response in an ethics of a job well done perspective, allowing them to maintain high patient safety standards regardless of the contingent situation for which safety first should be the motto of a disaster response plan.

## 1. Introduction

COVID-19 is an infectious disease caused by severe acute respiratory syndrome coronavirus 2 (SARS-CoV-2) [1]. The first COVID-19 cases were detected in China in December 2019 [2]. The WHO, after assessing the severity levels and global spread of the infection, declared on 11 March 2020 that the outbreak of COVID-19 was a pandemic [3]. Since the start of the pandemic, over 560 million people worldwide have been infected and over 6 million have died [4].

The COVID-19 pandemic was not only a public health issue, but also had a significant socio-economic impact by increasing poverty and inequality on a global scale [5].

For these reasons, the development and administration of vaccines against COVID-19 was a key element in the fight against the pandemic, as it protected health systems and helped restore global economies [6].

To date, there are 12,409,086,286 doses of vaccine administered in the world, 5,330,599,370 persons vaccinated with at least one dose, and 4,867,565,350 persons are fully vaccinated [4]. 

In Italy, the compulsory vaccination initially reserved for some categories of workers has been extended to all persons over the age of 50 regardless of the type of work activity [7,8]. A total of 95.54% of the population over 12 completed the primary vaccination cycle, 89.42% received the booster dose, and 23.49% received the second booster dose [9]. For healthcare workers (HWs), vaccination is a prerequisite for working and only in the presence of an established health hazard may it be postponed or excluded. HWs who do not wish to vaccinate themselves without valid reasons will not be allowed to practise their profession and will be assigned to other jobs or suspended without salary [10]. In this regard, it has been shown that independent predictors of HCW attitudes toward COVID-19 vaccination were the female sex, age >30 years, physician occupation, and unsure and positive attitudes about vaccination [11].

The development of national implementation plans and COVID-19 vaccination strategies ensured the immunization of large segments of the population in the shortest time. However, even before the start of the vaccination campaign, it was clear to decision-makers that the usual methods of vaccination were not suitable [12]. The pre-COVID-19 National Vaccine Prevention Plan focused on local health care units and general practitioners [13].

To face and overcome the challenge of a global public health emergency, it was necessary to provide specific staff and expertise, plan mass interventions, ensure the integration of software systems to handle the high volume of patient enrolment, organize locations for physical observation 15–30 min after vaccine administration and treatment of acute anaphylaxis, and set up extreme/standard cold chain and security against theft [14].

In Italy, in the initial phase of the vaccination campaign, hospital and peri-hospital sites were identified as well as mobile units for the immunization of people unable to reach the vaccination points [15]. In order to strengthen the existing vaccination network, public–private cooperation was also planned in a second phase, with the option of setting up vaccination centers at production sites, large-scale retail trade, gyms, schools, association or Episcopal Conference of Italy (CEI) facilities was envisaged [16].

Campus Bio-Medico University Hospital Foundation (CBM) is a teaching hospital that has been involved in the treatment of COVID-19 patients since the beginning of the pandemic. Therefore, the hospital underwent a reorganisation and reconversion of its spaces and activities to deal with this emergency.

The aim of this report is to share our experience in the organisation of spaces and activities of healthcare workers (HWs) to realise a fast, safe, and quality vaccination campaign.

In fact, on the one hand, the health organisations gave an excellent response to the largest vaccination campaigns in the history of medicine [17]. On the other hand, the rapidity and severity with which COVID-19 reduced patient safety suggests that the healthcare systems themselves did not have a sufficiently resilient safety infrastructure culture [18]. The negative impact COVID-19 had on healthcare-acquired infections (HAIs) rates and clusters of infections within hospitals [19] or central-line-associated bloodstream infections [20] are some examples.

The time is ripe [21]. Starting with an in-depth analysis of how the vaccination campaign was organised at our teaching hospital, we would like to emphasize a few lessons. Our experience compared to that of other hospitals could be useful to understand the strengths and weaknesses learnt from this experience and plan an effective, efficient, and resilient response to future pandemics right away.

## 2. Case Study

On December 2020, the CBM joined the COVID-19 vaccination campaign organized by the regional health department, as a hub center for the south-east area of Rome.

In two weeks, an interdisciplinary task force created a vaccination center (VC) and developed a vaccination program using plan–do–check–act (PDCA) cycles, to ensure high levels of quality and safety despite the high patient flows and the speed required to activate the process.

The task force, led by the hospital CEO, was composed by the medical director, the nursing director, the director of pharmacy, the head of the technical area, and the head of the administrative staff. In addition, to meet this challenge, the task force made use of experts in public health, informatics, hospital quality, and clinical risk management, who collaborated in the writing of the operational protocol and in the organization and implementation of the process.

The vaccination campaign was launched following the regional health department indications, starting with frail patients and then reaching the entire population. Based on an estimate of the population to be vaccinated, the regional health department defined the number and position of the VCs (HUBs) to be activated. Therefore, the offer was modulated based on the demand for vaccinations. The CVs were entered in a single IT portal of the regional health department, through which the population was able to book the slots for vaccination according to their preferences and the area they belong to.

Since December 2020, healthcare professionals and hospital staff most exposed to the risk of contagion (e.g., housekeeping staff and meal delivery staff) have been vaccinated. Therefore, on January 2021, the vaccination campaign for the entire population began, starting with the population group over the age of 80 and vulnerable people. Afterwards, the whole population have been gradually included in the vaccination schedule. Inpatients were also vaccinated where appropriate. From the opening of the VC, on 28 December 2020, until 11 August 2022, during 555 days of VC activity, 279,056 doses were administered, of which 4858 for health care professionals and most exposed staff of the hospital. The number of vaccines administered reached a daily maximum of 1543 doses with a mean of 503 doses per day. (Table 1 and Table 2).

The maximum number of vaccines that were given in one day was sometimes influenced by the number of doses supplied by the regional health department, not by the VC organization.

### 2.1. Organization of the Vaccination Center (VC) Building

To set up the VC, several important organizational aspects have been considered, based on the available evidences [22,23] and using the Center for Disease Control and Prevention (CDC) guidelines [24], which have been developed for a largescale influenza vaccination campaign. The CDC guidelines suggest structuring the entire flow in four main steps. The first is orientation, as people should be guided when arriving at the center, and they should be screened for potential symptoms; the second provides the filling of forms and the medical assessment; the third step is the vaccination; and the fourth step is the observation post-vaccination [23]. Therefore, the VC set-up was based on these guidelines. Built in two weeks and opened on 28 December 2020, the VC covers an area of 450 mt^2^ and it is equipped with an admission and waiting area, a pre-vaccination area, a vaccination area with eight examination rooms and eight stations for doses administration, and two large shock rooms for the monitoring of patients after vaccine administration. (Figure 1).

An administrative operator carries out the screening in the entrance by measuring the temperature and filling in the appropriate forms. Patients are then welcomed in the ‘Administration and waiting area’ by volunteers who support them in filling in the required forms and managing the incoming flow. Once the forms have been completed, they are led to the pre-vaccination area where they await the medical examination. Once completed the medical visit, they enter the vaccination room directly and remain in observation in the shock room for 30 min after the doses administration.

### 2.2. Staff Recruitment and Training

Professionals of the VC were recruited in a very short time and specially trained. In addition to the issues more strictly related to the management and organization of spaces and processes, specific training goals have been planned in order to ensure the best possible service for population. Staff were involved in a compulsory training program about vaccines management, temporary and definitive contraindications, suspicions of allergic reactions and emergency management, and also for data management and software use [12,22].

Moreover, targeted policies and procedures in force at the CBM Hospital have been disseminated, and compliance with the International Patient Safety Goals [23] was monitored in order to ensure high levels of quality and safety for patients.

The most relevant issues considered for patient safety and quality of care were those regarding patient identification, handover communication, the management of a high level of attention drugs, patient–professional communication, informed consent, and the management of clinical emergencies. In addition, professionals were also trained on how to report any clinical adverse events or near misses that might have occurred during their work shift.

Since its opening, the VC was active seven days a week—with the exception of some short periods of closure based on the regional health department indications and demand for vaccinations—with work shifts of doctors, nurses, and administrative staff organized in order to cover a schedule time comprised between 07:30 a.m. and 00:00 p.m., and to ensure the administration of the greatest possible number of doses even during periods of large influx, based on the number of reservations scheduled and the available doses.

The daily presence of professionals was organized so that for each available vaccine slot (chair) the presence of one doctor, one nurse, and one administrator was foreseen. The number of professionals present was variable and proportional to the slots booked every day in order to optimize the required resources. In addition, in order to facilitate the flow of people entering and leaving the VC, and to manage any critical issues, two volunteers and a guard were always present for each shift.

### 2.3. Vaccines Procurement, Storage, and Transport

The plan for the procurement, transport, preparation, and administration of vaccines, as well as the management of vaccine agendas for healthcare professionals, inpatients, and for the whole population, played a key role to ensure the success of the vaccination campaign [24]. The procurement of vaccines was organized through a centralized structure, guided by the pharmaceutical office of the regional health department. Vaccine bookings and registrations were managed by using a dedicated regional logistics software. For the conservation of vaccines, the pharmacy of CBM acquired four ultra-low freezers at −80 °C and four ultra-freezers at −20 °C, whose temperature control has been remotely and constantly monitored by the technical control room of the hospital, which is active 24 h a day. The following type of vaccines were administered at the CV: Pfizer, Moderna, Novavax, and AstraZeneca.

### 2.4. Vaccine Preparation and Administration

The task force defined and implemented a procedure for vaccines thawing, transport, storage, preparation, and administration, and for the management of any possible adverse drug reactions. Over the months, following suggestions of healthcare professionals and two reported near misses, the procedure was reviewed and subsequently improved. The involvement of the staff in continuous improvement proved to be an important leverage for an active and proactive risk management and to ensure safety for both patients and professionals [25]. In addition, to monitor the compliance with the procedure by the staff, a grid was used consisting of 33 items corresponding to each task of the process, and relating to the four main phases: thawing, preparation for dilution, dilution, and administration.

In order to increase safety for patients, a dedicated path was created for patients with allergies declared during the medical anamnesis. In particular, allergic patients were vaccinated in an outpatient clinic located within the university hospital, in order to facilitate the management of any possible clinical urgency. This approach ensured that the cases of adverse reactions were mild and easily manageable by the healthcare staff present in the VC.

### 2.5. Challenges and Opportunities

The components of the task force confirmed that the experience of opening a VC during the pandemic was a great challenge, but at the same time it allowed to discover new potentials and opportunities. Indeed, professionals reported motivation and the desire to work hard as part of the project, despite the effort and the fatigue. Actually, it often occurs that healthcare operators during emergencies [26] show greater resilience, dedication, and active cooperation to overcome difficulties and promote the common good. Many professionals involved in pandemic management reported a unique experience in terms of multiprofessional collaboration and the management of new and complex clinical and organizational situations [27]. This phenomenon can represent an important lever for change and growth [28].

At the same time, an emergency such as the COVID-19 pandemic increased the risk to patient safety exponentially, and it requires new strategies and new integrated approaches. Vaccinating the entire population requires mass outreach, but in many countries, such outreach is not a primary care capability [14].

An unprecedented volume of patient enrollment and scheduling was required, and it resulted in the opening of new spaces and structures in a few days.

The task force of CBM reports having experienced great flexibility in continuously updating on the basis of ministerial communications, and in quickly fulfilling the regional decisions aimed at promptly addressing the pandemic, and that one of the main challenges has been finding professionals available to guarantee a service is opened seven days a week.

In addition, other minor points of attention noted by frontline operators were the risks related to managing, simultaneously, different types of vaccines with different dosages, posology, storage methods, and stability, and the complexity linked to the vaccination of foreign people.

## 3. Discussion

The COVID-19 vaccines pose a special challenge for healthcare facilities. Hospitals’ managements involved in the opening of vaccination centers for COVID-19 had to face the effort to create and improve vaccine access, in particular for the most vulnerable people and for healthcare professionals involved in the frontline.

The management of vaccines and their doses, the limited availability of healthcare professionals, and the reorganization of spaces and logistics were only some of the main challenges hospitals had to deal with in order to reach the vaccination program in the shortest time and ensuring a high level of safety for the largest possible number of people [29].

To face this demanding program, many health facilities have placed all human, material, and intellectual resources to ensure high levels of quality and safety despite the high patient flows and speed required to activate the process.

Healthcare risk management is defined by the clinical and administrative activities undertaken to identify, assess, and reduce the risk of injury to patients, staff, and visitors and the risk of loss to the organisation itself [30].

Clinical risk management (CRM) aims to improve the quality and safe delivery of healthcare through procedures to identify and prevent circumstances that could put a patient at risk of an adverse event [31].

COVID-19 had a significant impact on the safety of care [32]. The pandemic drastically changed the provision of health services, particularly for hospital emergency departments, inpatient units, and in the organization of places to administer vaccines to large numbers of patients in the shortest time possible.

Therefore, although pandemic management and the care of patients with COVID-19 were prioritized, hospital adverse events were also influenced by unique situational factors [33]. Reports concerning policies and procedures predominated over classic clinical risk management issues such as falls, pressure injuries, and drug-related adverse events [34]. During the acute phase of the pandemic, it was suggested that CRM units support frontline operators, collect incident reports also through instant messaging tools, and analyse them within the specific task forces set up to deal with the health emergency [35].

However, despite public health policies and the efforts of the HWs, especially the first phase of the pandemic was characterised by widespread unpreparedness (incompetence), i.e., the inappropriateness of the structures and organisations in place to manage a probable pandemic [36]. It was foreseeable that there would be a pandemic, that we would need better antiviral agents, more effective flu vaccines, greater production capacity, and faster production speed [37]. It was likewise known that “the world is ill-prepared for a major pandemic or any other equally global, sustained, and threatening public health emergency” [38].

For the future, it will be mandatory to prepare public health action strategies in preparation for exceptional conditions with a transparent chain of responsibilities [39]. It also applies to that specific area of public health and legal medicine that aims at patient safety and improving the quality of healthcare.

To improve safety and the quality of care, health systems will have to promote legislative measures to enhance the clinical governance of health organisations, expand multisector and multinational synergies, and promote the involvement of patients and their families [40]. It will also be needed to capitalise on the relationships between patient safety and other priority issues such as climate safety, HWs safety and resilience, and patient reported experiences [41].

Health organisations should consider a model of health in which bio-social linkages resulting from the relationships between health conditions and socio-economic, political, and cultural determinants are emphasised.

To achieve this goal, the integration of natural and social science methods according to a systemic orientation will be necessary. Thus, for CRM there will be a pre-COVID-19 era and a post-COVID-19 era.

Indeed, the theoretical foundation of a systemic approach to CRM has its roots at the beginning of the last century, when science evolved through paradigm shifts that enabled a different way of seeing the world and thus of interpreting reality [42]. The failure of Hilbertian formalism resulted in the transformation of logic from linear to non-linear [43]. At the same time, Gödel’s incompleteness theorems [44], Heisenberg’s uncertainty principle [45], and Einstein’s theory of relativity [46] brought about a paradigm shift from Newtonian physics to complexity theory.

Based on this paradigm shift and in accordance with the objectives set out by the WHO [38] and the OECD [39], clinical risk management requires a complex view of reality permeated by a multidimensional perspective.

If the beginning of the 20th century was dominated by a shift in the scientific paradigm, the beginning of the 21st century could be characterised by a health care system distinguished by the increasing use of tools such as artificial intelligence (AI), internet of things (IoT) and big data [47]. In this scenario, enabling technologies can support CRM interventions according to a complex and multidimensional risk logic based on non-linear causal relationships and multi-causal effects.

However, it is required to use big data, not limiting its use to the correlation of data but also to its interpretation. The correlation between multifactorial e-tech data and the human interpretation of them is what we call systemic clinical risk management (SCRM) [48]. Adopting complexity theory and recognising the multi-causality of effect will allow those concerned with patient safety and quality of care to move from a mono-causal (cause–effect) dynamic to a multi-causal (cause–effect) dynamic that will need to be interpreted with the use of enabling technologies.

This may be the answer to the concept of “diffuse medicine” that the pandemic has highlighted, i.e., that one cannot make medicine from medicine alone, but every reality with which man is confronted can have an effect on the state of health [49].

The SCRM configures a proactive approach to clinical risk management based on the collection and processing of quantitative and qualitative data capable of developing a capacity to observe and summarise different elements in line with the multi-dimensional model suggested by the WHO [38] and the OECD [39].

SCRM has an intrinsic ethical component. A key factor in ethical evaluation is voluntariness, whereby responsibility in the health sector cannot be reduced to formal compliance with rules, procedures, and sanctions [50]. The ethical framework to which we refer is that of the ethics of job well done, which is embedded within ethical personalism [51]. According to the ethics of job well done, CRM should be characterised by co-working supported by realist knowledge, radical procedural innovation, the motivational involvement of all stakeholders, and an awareness that every medical act has an intrinsic ethical value [49].

Regarding complexity theory and systems thinking, the task force for the organization of the VC formed on an interdisciplinary co-design model. The expertise of professionals working in the fields of health care management, public health, nursing sciences, CRM, hospital quality, drug dispensing, distribution, management, and storage, management and design of technological facilities, information technology (IT), and administrative staff organization were integrated.

The establishment and organization of the VC was based on the best available scientific evidence [23,24,25,26]. Thus, the task force based its decision-making on realistic knowledge that always starts from experience and leads to seeking scientific truth as the foundation of its choices.

VC workers attended mandatory training programs covering vaccine management, temporary and definitive contraindications, suspected allergic reactions and emergency management, quality and patient safety, information management, and the use of IT systems. This made a motivational involvement of all workers in the VC and the recovery of the political dimension of job well done, that is, of professional excellence as a means of service to society and the common good. The original procedure was improved and implemented following suggestions from HCWs and two reports of near miss.

SCRM based on the ethics of a job well done has also proved its worth in an unusual condition characterized by a pandemic, high patient flows, and the need to activate a process quickly.

Putting the person at the *core* of the job has improved effectiveness and efficiency and ensured sustainability in the knowledge that every health act is a free and responsible human act with an intrinsic ethical value.

The COVID-19 vaccination campaign is difficult to compare with previous experiences. It was characterized by peculiar aspects such as the need to respect non-pharmaceutical interventions (NPIs) during administration procedures, a high precision in vaccine reconstitution by highly qualified personnel, extreme cold storage for mRNA vaccines, and the need to provide information to a large group of individuals [26].

Our experience could be discussed and integrated with other similar ones in order to achieve an effective, efficient, and resilient model. It has been shown that a successful hospital immunisation programme should consider five key elements: a multidisciplinary team that holds regular meetings, the use of automated IT tools, the use of data collection, review and dissemination tools, education and communication programmes, and the involvement of hospital leadership and patients [52]. In addition, it could also be considered in the context of a public health macro-perspective. It has been shown that in deciding where to locate health centres, policy makers should consider issues such as cost, accessibility, demand, and equity in order to minimise the impact of a pandemic [53].

## 4. Conclusions

COVID-19 had a significant negative impact on the safety of care [19,20,32].

There is therefore a need to re-evaluate health security with the aim of building a more resilient health care delivery system capable of maintaining high levels of safety even in times of crisis [18].

Healthcare-acquired infections (HAIs) rates and clusters of infections within hospitals [19], central-line-associated bloodstream infections [20], and incident reports related to policies and procedures [34] are examples of this.

The WHO and the OECD have shown that to implement a safety of care and a quality of health care, it is essential to consider the relationships between health and seemingly unrelated elements such as socio-economic, political, and cultural factors [42,43].

In this scenario, the correlation between multifactorial e-tech data obtained from enabling technologies and the human interpretation of the same according to a systemic approach can develop a capacity to observe and summarise different items in line with a multidimensional model. The adoption of a SCRM could guarantee healthcare organizations a more adequate and resilient response, allowing them to maintain high patient safety standards regardless of the contingent situation for which safety first should be the motto of a disaster response plan [54].

## Figures and Tables

**Figure 1 vaccines-10-01495-f001:**
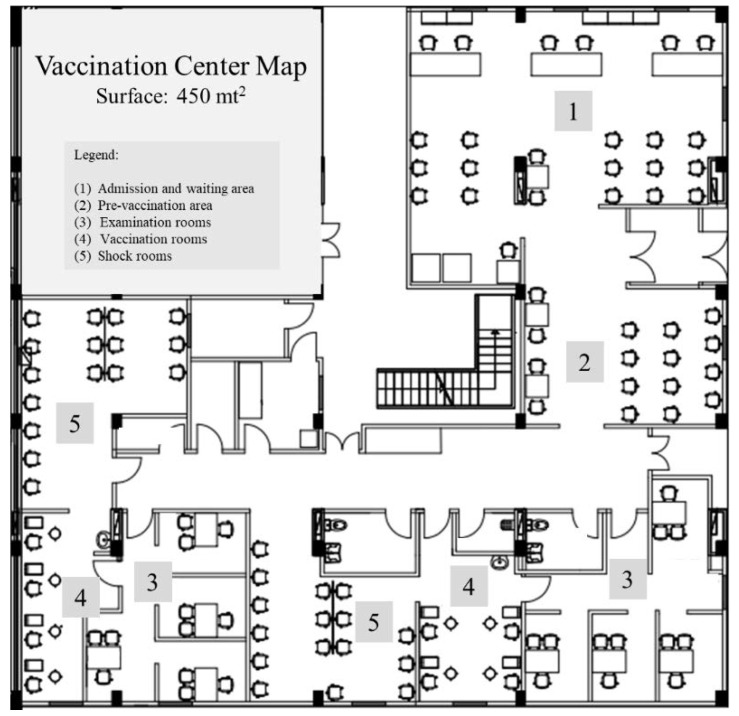
Vaccination Center (VC) map.

**Table 1 vaccines-10-01495-t001:** Details of the number of doses administered.

Given Doses	N.
Total doses administered	279,056
Mean per day	503
Max. per day	1543
Min. per day	3

**Table 2 vaccines-10-01495-t002:** Details of the number of doses administered per booster.

Given Doses	N.
First doses	100,360
Second doses	96,199
Third doses	74,981
Fourth doses	7516

## Data Availability

Not applicable.

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
