# Peer review of "Vaccines Administration in the Perspective of Patient Safety and Quality of Healthcare: Lesson from the Experience of an Italian Teaching Hospital for Pandemic Preparedness"

_vaccines, 2022, doi:10.3390/vaccines10091495_

Round 1
Reviewer 1 Report
Estimated Authors of the paper
"Vaccines administration in the perspective of patient safety and quality of healthcare: lesson from the experience of an Italian teaching hospital for pandemic preparedness"
I've read your report with great interest. In fact, in this paper, the experience on the vaccination centre of the Campus Biomedico of Rome has been reported, detailed and then summarized in order to stress the potential pros and cons of this very interesting topic (i.e. setting up and managing a mass vaccination centre).
Unfortunately, I'm unable to recommend the acceptance of this paper (in its current status) for the following reasons.
First of all, the present paper does represent a "case study", and the usual organization in introduction, materials and methods results and discussion fails to fit the aims and the content. A more appropriate organization in "introduction", "case study", "discussion", "conclusion" would be more accurate and would also improve the overall flow of this report.
Second, as for an original contribution, the present paper seemly lacks some comparison to other experiences on mass vaccination centres. Even though COVID-19 was an unprecedented event, mass vaccination centres were not: at the beginning of the vaccination campaign, Gianfredi et al. (https://pubmed.ncbi.nlm.nih.gov/34205891/) performed on this topic a rapid review that could provide some references for comparisons and discussion.
Third, actual figures are lacking. More precisely, according to the authors:
"From the opening of the Center until June 2022, more than 278.121 doses were administered, of which 4.858 for health care professionals and most exposed staff of the Hospital"
but, what about:
- how was this vaccination centre enclosed in the circle of vaccination centres of Rome and nearby areas? how many people were invited to this centre and following which rationale (i.e. residence, specific characteristics of the patients, its age...)
- the total number of daily doses (average and range)
- daily influx of people
- daily presence of professionals by occupational category (i.e. physicians, nurses, administrative personnel...)
- the number of cases requiring first aid and advanced life support because of side effects of vaccines
Such information should be provided and (hopefully) summarized in a series of specifically designed tables.
Author Response
Dear Reviewer #1,
all authors are grateful to you for considering our manuscript interesting and for your suggestions.
We accepted your proposals and implemented our manuscript.
- Reply to issues 1
The paper was reorganised. As recommended, the sections are: 'Introduction', 'Case study', 'Discussion', and 'Conclusion'.
- Reply to issues 2
As far as possible, we made a comparison with other vaccination experiences. Although, at present, it is not easy to compare COVID-19 vaccination with previous vaccination experiences for two reasons: the specificity of COVID-19 vaccination (compared to H1N1 influenza, smallpox, polyomyelitis, and typhoid fever); our case study concerns a mass vaccination centre built within a teaching hospital.
The following sentences and the corresponding references have been added: “The COVID-19 vaccination campaign is difficult to compare with previous experi-ences. It was characterized by peculiar aspects such as the need to respect non-pharmaceutical interventions (NPIs) during administration procedures, high preci-sion in vaccine reconstitution by highly qualified personnel, extreme cold storage for mRNA vaccines, the need to provide information to a large group of individuals [Gianfredi, 2021]. Our experience could be discussed and integrated with other similar ones in order to achieve an effective, efficient and resilient model. It has been shown that a successful hos-pital immunisation programme should consider five key-elements: a multidisciplinary team that holds regular meetings, the use of automated IT tools, the use of data collection, review and dissemination tools, education and communication programmes, involve-ment of hospital leadership and patients [Hofstetter, 2022]. In addition, it could also be considered in the context of a public health macro-perspective. it has been shown that in deciding where to locate health centres, policy makers should consider issues such as cost, accessibility, demand and equity in order to minimise the impact of a pandemic [Delgado, 2022] (Gianfredi, V.; Pennisi, F.; Lume, A.; Ricciardi, G.E.; Minerva, M.; Riccò, M.; Odone, A.; Signorelli, C. Challenges and Opportunities of Mass Vaccination Centers in COVID-19 Times: A Rapid Review of Literature. Vaccines 2021, 9, 574; Hofstetter, A. M.; Rao, S.; Jhaveri, R.. Beyond Influenza Vaccination: Expanding Infrastructure for Hospital-based Pediatric COVID-19 Vaccine Delivery. Clin Ther 2022, 44, 450–455; Delgado, E.J.; Cabezas, X.; Martin-Barreiro, C.; Leiva, V.; Rojas, F. An Equity-Based Optimization Model to Solve the Location Problem for Healthcare Centers Applied to Hospital Beds and COVID-19 Vaccination. Mathematics 2022, 10, 1825)
- Reply to issues 3
The paper was improved, adding the information required. As recommended, the sections are: 'Introduction', 'Case study', 'Discussion', and 'Conclusion'.
The par. 2 has been improved adding/reviewing the following sentences and adding table 1 and table 2:
“The vaccination campaign was launched following the regional health department indications, starting with frail patients and then reaching the entire population. Based on an estimate of the population to be vaccinated, the regional health department defined the number and position of the VCs (HUBs) to be activated. Therefore, the offer was modulated based on the demand for vaccinations. The CVs were entered in a single IT portal of the regional health department, through which the population was able to book the slots for vaccination according to their preferences and the area they belong to.
Since December 2020, healthcare professionals and Hospital staff most exposed to the risk of contagion (eg. housekeeping staff, meal delivery staff) have been vaccinated. Therefore, on January 2021 the vaccination campaign for the entire population began, starting with the population group over the age of 80 and vulnerable people. Afterwards, the whole population have been gradually included in the vaccination schedule. Inpatients were also vaccinated where appropriate. From the opening of the VC, on 28 December 2020, until 11 August 2022, during 555 days of VC activity, 279.056 doses were administered, of which 4.858 for health care professionals and most exposed staff of the Hospital. The number vaccines administered reached a daily maximum of 1.543 doses with a mean of 503 doses per day. (Table 1, Table 2)
The maximum number of vaccines that were given in one day was sometimes influenced by the number of doses supplied by the regional health department, not by the VC organization.”
Table 1. Details of the number of doses administered.
Given doses |
N. |
Total doses administered |
279.056 |
Mean per day |
503 |
Max. per day |
1.543 |
Min. per day |
3 |
Table 2. Details of the number of doses administered per booster.
Given doses |
N. |
First doses |
100.360 |
Second doses |
96.199 |
Third doses |
74.981 |
Fourth doses |
7.516 |
The par. 2.2 has been improved adding the following sentences:
“Since its opening, the VC was active seven days a week – with the exception of some short periods of closure based on the regional health depart-ment indications and demand for vaccinations - with work shifts of doctors, nurses and administrative staff organized in order to cover a schedule time comprised between 07:30 a.m. and 00:00 p.m., and to ensure the administration of the greatest possible number of doses even during periods of large influx, based on the number of reservations scheduled and available doses.
The daily presence of professionals was organized so that for each available vaccine slot (chair) the presence of one doctor, one nurse and one administrator was foreseen. The number of professionals present was variable and proportional to the slots booked every day in order to optimize the required resources. In addition, in order to facilitate the flow of people entering and leaving the VC, and to manage any critical issues, two volunteers and a guard were always present for each shift.”
The par. 2.4 has been improved adding the following sentences:
“In order to increase safety for patients, a dedicated path was created for patients with allergies declared during the medical anamnesis. In particular, allergic patients were vaccinated in an outpatient clinic located within the University Hospital, in order to facilitate the management of the any possible clinical urgency. This approach ensured that the cases of adverse reactions were mild and easily manageable by the healthcare staff present in the VC.”
- Below are the added references
- Delgado, E.J.; Cabezas, X.; Martin-Barreiro, C.; Leiva, V.; Rojas, F. An Equity-Based Optimization Model to Solve the Location Problem for Healthcare Centers Applied to Hospital Beds and COVID-19 Vaccination. Mathematics 2022, 10, 1825.
- Gianfredi, V.; Pennisi, F.; Lume, A.; Ricciardi, G.E.; Minerva, M.; Riccò, M.; Odone, A.; Signorelli, C. Challenges and Opportunities of Mass Vaccination Centers in COVID-19 Times: A Rapid Review of Literature. Vaccines 2021, 9, 574.
- Hofstetter, A. M.; Rao, S.; Jhaveri, R.. Beyond Influenza Vaccination: Expanding Infrastructure for Hospital-based Pediatric COVID-19 Vaccine Delivery. Clin Ther 2022, 44, 450–455.
- Ministero della Salute. Piano nazionale prevenzione vaccinale 2017-2019 (PNPV). Available online: https://www.salute.gov.it/portale/vaccinazioni/dettaglio ContenutiVaccinazioni.jsp? lingua=italiano&id=4828&area= vaccinazioni&menu= vuoto (accessed on 24 August 2022)
- Peruch, M.; Toscani, P.; Grassi, N.; Zamagni, G.; Monasta, L.; Radaelli, D.; Livieri, T.; Manfredi, A.; D’Errico, S. Did Italy Really Need Compulsory Vaccination against COVID-19 for Healthcare Workers? Results of a Survey in a Centre for Maternal and Child Health. Vaccines 2022, 10, 1293.
- Quintiliani, L.; Sisto, A.; Vicinanza, F.; Curcio, G.; Tambone, V. Resilience and psychological impact on Italian university students during COVID-19 pandemic. Distance learning and health. Psychol Health Med 2022, 27 69–80.
- Zdravkovic, M.; Popadic, V.; Nikolic, V.; Klasnja, S.; Brajkovic, M.; Manojlovic, A.; Nikolic, N.; Markovic-Denic, L. COVID-19 Vaccination Willingness and Vaccine Uptake among Healthcare Workers: A Single-Center Experi-ence. Vaccines 2022, 10, 500.
For details, please see the revised and updated file attached.
We hope that your comments have been met in this way.
We look forward to your reply.
Kind regards,
Dr. Anna De Benedictis, Ph.D
Corresponding author
Reviewer 2 Report
The manuscript “Vaccines administration in the perspective of patient safety and quality of healthcare: lesson from the experience of an Italian teaching hospital for pandemic preparedness” is an example of a deep analysis of the COVID-19 vaccination program. It is well explained that the organized vaccination campaign could be useful to understand strengths and weaknesses, learn from this experience, and plan an effective, efficient and resilient response to future pandemics. In my view, it should be published without any specific modification.
Author Response
Dear Reviewer #2,
All authors are grateful to you for considering our manuscript interesting and for approving its publication.
Kind regards,
Dr. Anna De Benedictis, Ph.D
Corresponding author
Reviewer 3 Report
Dear Authors,
my congratulations to this interesting and original manuscript!
Could you please provide information on the counts of professional personnel / volunteers occupied in the vaccination program (approximate total hours?).
I have nothing more to say about the manuscript other than recommending it for publication.
Best regards,
Author Response
Dear Reviewer #3,
all authors are grateful to you for considering our manuscript interesting and for your suggestions.
We accepted your proposals and added information on the number of professional/voluntary staff involved in the vaccination programme in the par. 2.2.
For details, please see the revised and updated file attached.
We hope that your comments have been met in this way.
We look forward to your reply.
Kind regards,
Dr. Anna De Benedictis, Ph.D
Corresponding author
Reviewer 4 Report
The interest of this article lies in the fact that it is based on clinical practice and can serve as a guideline for centers that will have to develop vaccination campaigns against covid19 or other future threats. There are many studies on the scientific aspects of vaccination, but few studies consider the practical and logistical aspects. It is simply a matter of writing and organizing the manuscript.
Criticism
1. Overall the article has two parts, empirical data, and a theoretical model. They should be better linked. There needs to be a smooth transition between the data from the center’s experience and the CRM and bioethical models they use. Now they are similar issues.
Part of the theoretical considerations on CRM should be moved to the discussion to introduce the reader to the theoretical framework.
2. Is this sentence correct? “5 billion people vaccinated with at least one dose and almost 5 billion.” Please check the numbers because it seems very coincidental that the figures are the same. (Lines 4-6)
3. Avoid the word billion because it generates confusion. In USA and UK, it is 1,000 million, while in Europe it is. 1,000,000 millions. (lines 44 to 45)
4. The sentence “To date, there are over 12 billion doses of vaccine administered in the world, more than 5 billion people vaccinated with at least one dose and almost 5 billion people are fully vaccinated.”
It should be written as follows
To date, there are over 12,000 million doses of vaccine administered in the world, more than 5,000 million people are vaccinated with at least one dose, and almost 5,000 million people are fully vaccinated
5. Please note that people from outside Italy are not familiar with the Italian health care system, nor with the organization of public health care in Italy.
Please explain in more detail to what extent vaccination is mandatory. What happens if a person is antivaccine or does not wish to be vaccinated for religious reasons? Authors could use this citation https://www.mdpi.com/2076-393X/10/8/1293
6. It is stated that before the start of the vaccination campaign, it was clear to decision-makers that the usual vaccination methods were not suitable. Please explain succinctly why.
7. In line 13, it is stated that Episcopal Conference of Italy (CEI) facilities were envisaged. The CEI is the owner of the facilities, and it should be stated what type of CEI facilities were used, e.g., churches, cathedrals, schools, and hospitals. The information about the facility owner is interesting because it can show the existence of a public-private partnership. However, the type of facilities or centers should be indicated.
8. In order to understand the logistical issues, it is necessary to indicate the type of vaccine and the brand used since each type of vaccine has different storage and thawing requirements.
9. Please include as supplementary material “the 33 items grid” mentioned in line 184.
. When talking about personalism, a quote should be given.
It is indicated in the text the maximum number of vaccines that were given in one day. It would be interesting to explain if factors limited the number and what they were. For example, the number of nurses at the vaccination posts and the number of posts in the shock room.
. On the other hand, the discussion should relate the theoretical framework they use to the vaccination performed in their hospital.
3. In the discussion, compare what was done in the Biomedical Hospital of Rome with the experience of other centers in Italy or the world.
The following references could be of interest
https://www.mdpi.com/2227-7390/10/11/1825
https://doi.org/10.3390/vaccines10040500
https://pubmed.ncbi.nlm.nih.gov/35172946/
Author Response
Dear Reviewer #4,
all authors are grateful to you for considering our manuscript interesting and for your suggestions.
We accepted your proposals and implemented our manuscript.
a. Reply to issues 1
We accepted the criticism and thank the reviewer for this remark. We have re-written and implemented the "Discussion" to make the theoretical framework more consistent with the experience of the vaccination centre.
b. Reply to issues 2, 3 and 4.
A data check was carried out and the word billion has been removed. The sentence has been rewritten: “To date, there are 12.409.086.286 doses of vaccine administered in the world, 5.330.599.370 persons vaccinated with at least one dose and 4.867.565.350 persons are fully vaccinated.”
c. Reply to issue no. 5
The following sentence and the corresponding reference have been added: “For healthcare workers (HWs), vaccination is a prerequisite for working and only in the presence of an established health hazard may it be postponed or excluded. HWs who do not wish to vaccinate themselves without valid reasons will not be allowed to practise their profession and will be assigned to other jobs or suspended without salary” (Peruch, M.; Toscani, P.; Grassi, N.; Zamagni, G.; Monasta, L.; Radaelli, D.; Livieri, T.; Manfredi, A.; D’Errico, S. Did Italy Really Need Compulsory Vaccination against COVID-19 for Healthcare Workers? Results of a Survey in a Centre for Maternal and Child Health. Vaccines 2022, 10, 1293. https://doi.org/10.3390/vaccines10081293)
d. Replay to issue no. 6
The following sentence and the corresponding reference have been added: “The pre-COVID-19 National Vaccine Prevention Plan focused on local health care units and general practioners (Ministero della Salute. Piano nazionale prevenzione vaccinale 2017-2019 (PNPV). Available online:https://www.salute.gov.it/portale/vaccinazioni/dettaglioContenutiVaccinazioni.jsp?lingua=italiano&id=4828&area=vaccinazioni&menu=vuoto (accessed on 24 August 2022)).”
e. Replay to issue no. 7
Unfortunately, we cannot specify which facilities of the Episcopal Conference of Italy (CEI) were involved. The Italian Government's Anti-Covid Vaccination Plan aimed to enhance the existing vaccination network through the involvement of production sites, large-scale retail trade, gyms, schools, associations or Episcopal Conference of Italy (CEI) facilities. It does not indicate exactly the type of facilities or centres. However, we found the consideration that a public-private partnership emerges from the manuscript very useful. For this reason we have emphasised it in the manuscript.
f. Replay to issue no. 8
We added in the par. 2.3 the type of vaccine and the brand used, which are: Pfizer, Moderna, Novavax and Astazeneca.
g. Replay to issue no. 9
- A reference for bioethical personalism has been added.
- It has been explained in the par. 2 if factors limited the number and what they were. The following sentence has benn added: “The maximum number of vaccines that were given in one day was sometimes influenced by the number of doses supplied by the regional health department, never by the VC organization.”
- We have re-written and implemented the "Discussion" to make the theoretical framework more consistent with the experience of the vaccination centre.
h. Replay to issue no. 10
We compared our experience with those proposed by the reviewer
i. Below are the added references
- Delgado, E.J.; Cabezas, X.; Martin-Barreiro, C.; Leiva, V.; Rojas, F. An Equity-Based Optimization Model to Solve the Location Problem for Healthcare Centers Applied to Hospital Beds and COVID-19 Vaccination. Mathematics 2022, 10, 1825.
- Gianfredi, V.; Pennisi, F.; Lume, A.; Ricciardi, G.E.; Minerva, M.; Riccò, M.; Odone, A.; Signorelli, C. Challenges and Opportunities of Mass Vaccination Centers in COVID-19 Times: A Rapid Review of Literature. Vaccines 2021, 9, 574.
- Hofstetter, A. M.; Rao, S.; Jhaveri, R.. Beyond Influenza Vaccination: Expanding Infrastructure for Hospital-based Pediatric COVID-19 Vaccine Delivery. Clin Ther 2022, 44, 450–455.
- Ministero della Salute. Piano nazionale prevenzione vaccinale 2017-2019 (PNPV). Available online: https://www.salute.gov.it/portale/vaccinazioni/dettaglio ContenutiVaccinazioni.jsp? lingua=italiano&id=4828&area= vaccinazioni&menu= vuoto (accessed on 24 August 2022)
- Peruch, M.; Toscani, P.; Grassi, N.; Zamagni, G.; Monasta, L.; Radaelli, D.; Livieri, T.; Manfredi, A.; D’Errico, S. Did Italy Really Need Compulsory Vaccination against COVID-19 for Healthcare Workers? Results of a Survey in a Centre for Maternal and Child Health. Vaccines 2022, 10, 1293.
- Quintiliani, L.; Sisto, A.; Vicinanza, F.; Curcio, G.; Tambone, V. Resilience and psychological impact on Italian university students during COVID-19 pandemic. Distance learning and health. Psychol Health Med 2022, 27 69–80.
- Zdravkovic, M.; Popadic, V.; Nikolic, V.; Klasnja, S.; Brajkovic, M.; Manojlovic, A.; Nikolic, N.; Markovic-Denic, L. COVID-19 Vaccination Willingness and Vaccine Uptake among Healthcare Workers: A Single-Center Experi-ence. Vaccines 2022, 10, 500.
For details, please see the revised and updated file attached.
We hope that your comments have been met in this way.
We look forward to your reply.
Kind regards,
Dr. Anna De Benedictis, Ph.D
Corresponding author
Round 2
Reviewer 1 Report
Estimated Authors,
my previous concerns have been accurately addressed; I've no further requests or suggestions, and I therefore endorse the acceptance of the present paper.